# Preliminary Investigation of Effect of Neem-Derived Pesticides on *Plasmopara halstedii* Pathotype 704 in Sunflower under *In Vitro* and *In Vivo* Conditions

**DOI:** 10.3390/plants9040535

**Published:** 2020-04-21

**Authors:** Pratik Doshi, Nisha Nisha, Ahmed Ibrahim Alrashid Yousif, Katalin Körösi, Rita Bán, György Turóczi

**Affiliations:** Department of Integrated Plant Protection, Plant Protection Institute, Faculty of Horticultural Sciences, Szent István University, Páter Károly utca 1, H-2100 Gödöllő, Hungary; nisha27evs@gmail.com (N.N.); ahmadalrashed45@gmail.com (A.I.A.Y.); Korosi.Katalin@mkk.szie.hu (K.K.); Ban.Rita@mkk.szie.hu (R.B.); Turoczi.Gyorgy@mkk.szie.hu (G.T.)

**Keywords:** *Plasmopara halstedii*, azadirachtin, neem leaf extract, sunflower, NeemAzal T/S, biological control

## Abstract

Two neem-derived pesticides were examined under *in vitro* and *in vivo* conditions to test their efficacy in controlling *Plasmopara halstedii* pathotype 704, a causal agent of downy mildew in sunflower. All the tested concentrations of neem leaf extract and the highest concentration of commercial neem product significantly reduced the sporangial germination under *in vitro* conditions. In *in vivo* experiment, 3-days old pre-treated seedlings with both concentrations of neem leaf extract and the highest concentration of commercial product showed a significant reduction in the infection indicating possible systemic effect of neem. When the seedlings were treated following the infection with *P. halstedii* (i.e., post-treatment), only the highest concentrations of neem leaf extract and the commercial product showed a significant reduction in the infection indicating curative effect of neem. Possibilities for the control of *P. halstedii* with neem-derived pesticides are discussed.

## 1. Introduction

Sunflower *Helianthus annuus* L. is one of the most important oilseed crops in the world. Downy mildew is a major disease caused by the biotrophic oomycete *Plasmopara halstedii* (Farl) Berlese et de Toni. This pathogen affects sunflower yield losses, more than 85%, as well as the cost of sunflower crop protection and resistance breeding [1,2,3]. From an economic and scientific point of view, the community of oomycete molecular geneticists ranked *P. halstedii* as the 16th most important oomycete [4,5]. The distribution and genetic variability of *P. halstedii* have been studied extensively, especially in North America and Europe.

This disease is mostly initiated by the soil-borne oospores and occasionally from infected seeds. In this context, the most susceptible stage of host development is between germination and emergence [6]. *Plasmopara halstedii* infection in the sunflower usually takes place in the below ground plant parts by direct penetration in the roots [7]. Systemic infection of sunflowers by *P. halstedii* occurs much more readily through hypocotyls than through roots [8]. Seedling damping-off occurs by root infection, whereas severe symptoms, such as stunted plants (dwarfing), chlorosis of leaves, and white sporulation to the production of infertile flowers, ultimately resulting in yield loss [9,10].

*Plasmopara halstedii* rapidly develop races (pathotypes) that can break down the resistance genes in sunflowers [11,12]. Since the pathogen shows high phenotypic diversity, particularly in virulence and fungicide sensitivity, effective disease control depends on a profound knowledge of the biology of the pathogen, its physiological capacities, and requirements, as well as the molecular mechanisms involved in the interaction with the host and environment [7].

Downy mildew of sunflower can be controlled by using resistant cultivars, agrotechnical methods, and chemical treatment (with fungicides) of the seeds with Mefenoxam [13]. However, the pathotypes have developed fungicide resistance and have overcome plant resistance genes (*PI* resistance genes) [10]. 

Widespread fungicide research has led to finding alternate solutions, which are sustainable, economical, and eco-friendly. Plant material and their respective by-products/compounds offer a great scope as an alternative to these chemicals [14]. Neem (*Azadirachta indica* A. Juss) plant protection products are known to possess antifeedant, antifungal [15,16], nematicidal, insecticidal [16] properties. Efficacy of seed, leaf, and bark extract of the neem tree is known worldwide for controlling insect pests and in agriculture [17]. Achimu and Schlösser [17] demonstrated the effect of neem seed extract and commercial neem product on *Plasmopara viticola* (Berk. and M.A. Curtis) Berl. and de Toni in grapevine and found that there was a protective treatment efficacy of over 90%.

Over the past decades, sunflower hybrids containing the *Pl6* resistance gene against sunflower downy mildew have grown worldwide. This gene confers resistance against many pathotypes of *P. halstedii*. As a result, the incidence of sunflower downy mildew has increased significantly in the main sunflower producing regions. Although hybrids with developed resistance genes against *P. halstedii* are available, alternative methods are of increasing importance in the integrated management to control this pathogen and its new pathotypes.

No previous records (or literature) are available about the effects of neem-derived pesticides on *P. halstedii* in sunflower. Therefore, the aim of this study is to investigate the effects of neem-derived pesticides, i.e., neem leaf extracts (containing more compounds in addition to azadirachtin) and a commercial neem-based pesticide product NeemAzal T/S (denoted as AZA) (containing only 1% azadirachtin) on *P. halstedii* under *in vitro* and *in vivo* conditions in sunflower. Different possible reasons for how the neem-derived pesticides work to control *P. halstedii* are discussed.

## 2. Results

### 2.1. HPLC Analysis of Neem Leaf Extract (NLE)

The chromatogram of the analysis of the standard azadirachtin can be seen in Figure 1 while the chromatogram of neem leaf extract sample is shown in Figure 2. The peak 1 in both spiked and non-spiked graphs is the azadirachtin A concentration, which can be verified when compared to the standard azadirachtin A, and which had the same spectrum as azadirachtin A. The peaks 2–5 in both the spiked and non-spiked samples (Figure 2) are thought to be the derivatives or isomers of azadirachtin. This was verified by comparing the spectrum of these peaks (2–5) with standard azadirachtin A peak, which was 213–214 nm.

Peak 1 from the spiked and non-spiked neem leaf extract sample was confirmed to be azadirachtin A when compared to the standard azadirachtin A, as they had the same spectrum (i.e., 213–214 nm) and retention time (Figure 2). There were other peaks, namely from 2 to 5, which appeared on the chromatogram for the neem leaf extract sample. They are suspected to be the derivatives of azadirachtin as they too appeared in the same spectrum as that of the standard. The concentrations of these peaks were calculated using the formula mentioned in the materials and methods for both spiked and non-spiked samples (Table 1). It can be further extrapolated that azadirachtin A concentration in 10 g and 20 g of neem leaf extract is 2 mg and 4 mg, respectively.

### 2.2. Pathotype Characterization of the Tested P. halstedii Isolate

The result of pathotype characterization of the *P. halstedii* isolate used in the test is shown in Table 2. Infection rates on 4 out of the 9 differential lines (Iregi szürke csíkos, RHA-265, RHA-274 and HA-335) were between 91.7% and 100% at the first evaluation, and as many as 100% at the second evaluation. Most of the infected plants in each differential line showed damping-off by the time of the second evaluation, so these lines were highly susceptible to the examined pathotype of sunflower downy mildew. Even differential line HA-335 containing resistance gene *Pl6* was not effective against the examined isolate. There were no infected plants among the rest of the differential lines, so these were considered as resistant to the *P. halstedii* isolate examined. Summing the scores given according to the reactions of the differential lines by each triplet, the examined *P. halstedii* isolate could be identified as pathotype 704. 

### 2.3. In Vitro Experiment: Microscopic Examination of the Effect of Neem-Derived Pesticides on P. halstedii Inoculum

The microscopical examination of sporangia was done 24 h after treatment with neem leaf extract and NeemAzal T/S (1% azadirachtin). The statistical analysis shows that all the neem-derived pesticide treatments showed a significant inhibition on the total number of empty sporangia (mean sq = 86.0, F value = 5.811, *p* < 0.05).

All the treatments, except AZA 0.01%, were found to be significantly better than the control (no treatment applied) at reducing the total number of empty sporangia (which includes completely empty and partially empty sporangia, as per the proposed scale; Figure 3). Both concentrations of neem leaf extract and the highest concentration of NeemAzal T/S gave similar results to those of Mefenoxam.

### 2.4. In Vivo Experiment: Pre- and Post-Treatment Effect of Neem-Derived Pesticides Against P. halstedii in Sunflower

*In vivo* experiment was performed using susceptible sunflower variety. In the case of pre-treatment, it is evident that both concentrations of neem leaf extract and the highest concentration of NeemAzal T/S were found to significantly reduce the sporulation along with Mefenoxam, while the lowest concentration of NeemAzal T/S (AZA 0.01%) was not significant in reducing the sporulation as compared to control (Figure 4). The pre-treatment had a significant effect on the sporulation (Table 3).

In the case of post-treatment, the lowest concentration of both neem leaf extract and NeemAzal T/S did not reduce the sporulation and damping-off significantly as compared to control (Figure 4). However, the highest concentration of both neem leaf extract and NeemAzal T/S significantly reduced the sporulation and the number of damped-off seedlings. The lowest sporulation and damping-off were observed in the Mefenoxam treatment and was significantly different from all the other treatments. The statistical results of ANOVA showed that both pre- and post-treatment of infected seedlings using neem-derived pesticides had a significant effect on the sporulation under *in vivo* condition (Table 4). The mean square sporulation in the pre-treatment was found to be lower compared to post-treatment (Table 4).

For initial plant height (10 days after sowing), in the case of pre-treatment, plant height measured for AZA 0.01% was significantly shorter compared to other treatments of neem leaf extracts, AZA 0.1%, and Mefenoxam, but was not significantly different from the infected control (seed plus inoculum). Plants pre-treated with different concentrations of neem leaf extracts, AZA 0.1%, and Mefenoxam and then inoculated with *P. halstedii* showed no significant difference to plants treated with bidistilled water (Figure 5).

In the case for post-treatment (Figure 5), the lowest concentrations, i.e., NLE 10% and AZA 0.01% showed significantly shorter plant height as compared to Mefenoxam and plants treated with bidistilled water. None of the treatments showed any significant difference in the plant height compared to the infected control (i.e., seed plus inoculum). 

No chlorosis was observed in any of the plants inoculated with *P. halstedii* after a period of 19 days (i.e., at the end of the experiment). 

## 3. Discussion

It is evident from our result that the azadirachtin A content in the leaves is very low (0.2 µg/g = 1 mg/5 g leaves). The same result was observed by Ghimeray et al. [18], where they found trace amounts of azadirachtin content in the water extract in the neem leaves grown in the foothills of Nepal. However, it has been reported that azadirachtin is highly concentrated in mature seeds [19]. Other peaks were detected on the same spectrum as that of azadirachtin, but the retention time was different as compared to azadirachtin A. It is suspected that these compounds are potential derivatives of azadirachtin A. Our potential azadirachtin derivative suspects can be confirmed by Kumar et al. [19], as they stated that different analogues of azadirachtin A having similar biological activity has been identified to be azadirachtin B-L.

The effect of two different neem-derived pesticides, namely neem leaf extract and NeemAzal T/S, was tested against *Plasmopara halstedii*, the causal agent of sunflower downy mildew *in vitro* and *in vivo* conditions. Under *in vitro* conditions, all the treatments, except the lowest concentration of NeemAzal T/S, showed a significant reduction in the sporangial germination. Our results are similar to those reported by Mirza et al. [20] where they tested different neem products against different stages of an oomycete, *Phytophthora infestans* (Mont.) de Bary (*P. infestans*). They found that all the neem products, namely crude neem seed oil, nimbokil (a commercial formulation of neem oil), crude neem seed oil terpenoid extract, and neem leaf decoction were effective against mycelial growth, sporangial germination, and sporangium production of *P. infestans*. Rashid et al. [14] observed similar efficacy when they compared different neem products against two isolates of *P. infestans*. They found that all the neem products tested significantly inhibited the different developmental stages of this oomycete. Our results are also consistent with Mboussi et al. [21] after they tested the effects of aqueous extract of neem and *Thevetia peruviana* seeds (known for antifungal properties) on *P. infestans,* and concluded that both extracts had the same effects as Ridomil Gold Plus, a chemical fungicide with effective ingredients Mefenoxam and copper, against *Phytophthora megakarya,* Brasier and M.J. Griffin. The results of this experiment are also in line with previous reports of Ngadze [22], where *Azadirachta indica* (Neem) was found to be effective against *P. infestans* both under *in vitro* and *in vivo* conditions. 

During *in vivo* tests, in the pre-treatment experiment, we found in our study that both the selected concentrations of neem leaf extract and highest concentration of NeemAzal T/S successfully reduced the infection in sunflower. Our results contradict the findings of Rovesti et al. [23], where neem extract was found to be ineffective against *P. infestans,* but they are in line with the results found by Ngadze [22] where an *in vivo* experiment suggested that extracts of both *Allium sativum* and neem were effective to control *P. infestans* in potatoes. Our results were also consistent with Achimu and Schlösser [17], where neem seed extract and commercial neem products were highly effective against *Plasmopara viticola* (Berk. and M.A. Curtis) Berl. and De Toni, 1888, in grapevine. Moreover, Krzyzaniak et al. [24] found the same results as Achimu and Schlösser [17], where they successfully controlled *P. viticola* using plant extract. It can be said that these results might be due to the presence of different biologically active compounds such as azadirachtin in neem leaves and other plant parts. Shakywar et al. [25] also found similar results under *in vivo* conditions and they stated that the inhibitory action in the neem leaf extract might be due to azadirachtin present in all parts of the plant. The reduction of the infection in the pre-treatment might be the result of the sensitizing of sunflower defense response towards *P. halstedii,* which was also reported by Fernandez et al. [26], where they tested the essential oil obtained from *Bupleurum gibraltarium* against *P. halstedii*. They reported that the oil pre-treatment might activate the defense response of the seedlings against *P. halstedii.*


One of the possible reasons of controlling *P. halstedii* infection could be the systemic effect of neem. Systemic effect of neem is validated by various studies conducted by Naumann et al. [27] who found that the mountain pine beetle population in Lodgepole pine was reduced due to upward translocation of azadirachtin. Similar results were observed by Osman and Port [28] and Marion et al. [29], where they observed translocation of azadirachtin in vegetable crops and in birch (*Betula* spp.), respectively. Goel et al. [30] demonstrated the systemic acquired resistance in tomato induced by neem fruit extracts against bacterial speak cause by *Pseudomonas syringae* pv. tomato and Bhuvaneshwari et al. [31] demonstrated the same systemic effect in barley seedlings against *Drechslera graminea*. 

In the case of post-treatment, highest concentrations of both the neem-derived pesticides inhibited the infection. This might be attributed to the curative effect of neem-derived pesticides. The curative effect of neem products was observed by Achimu and Schlösser [17], who stated that inhibition of indirect germination of sporangia, by preventing zoospore release and/or formation, explains the efficacy of these products, which can be validated through the *in vitro* results of this experiment. Perhaps azadirachtin in the neem leaf extract alone may or may not cause this inhibitory action. There might be more than one biologically active compound working synergistically to control the infection that are different from azadirachtin and related substances [32,33]. 

Plant extracts possessing different properties against pest and pathogens can prove to be beneficial where chemical pesticides fail; hence, a thorough and extensive research in this field is needed. This is the first report of neem leaf extracts and commercially available azadirachtin exhibiting strong antifungal activity against *P. halstedii*. It is a naturally available fungicide (the neem tree) and a promising alternative to chemical pesticides for controlling downy mildew in sunflower by whole seedling treatment.

## 4. Conclusions

The effects of neem-derived pesticides on *P. halstedii* pathotype 704 under *in vitro* and *in vivo* conditions were examined for the first time through this research. Our results proved that neem-derived pesticides can be a valuable fungicide in controlling downy mildew of sunflower. Although it is the first step towards testing neem-derived pesticides against *P. halstedii,* further research is needed to test the effect of neem extracts and neem-derived commercial products on different pathotypes of *P. halstedii* under **in vitro**, glasshouse, and field conditions with different modes of application. In addition, an investigation on the systemic and/or curative effect of neem-derived pesticides against *P. halstedii* in sunflower by measuring different enzymatic activities in the plants needs to be done. It is highly recommended to test the freshly harvested seeds from the field that were previously treated with neem-derived pesticides to check the presence of different biologically active compounds of neem in them. 

## 5. Materials and Methods

### 5.1. Preparation of Neem Leaf Extract (NLE)

Air-dried neem leaves were obtained from the local market of Mumbai, India. The leaves were ground into powder using an electric blender. The methodology for preparing neem leaf extract was followed according to Doshi et al. [34] and Petrikovszki et al. [35] with slight modifications. Two concentrations of 10% and 20% (*w/v*) were prepared by suspending 10 g and 20 g of neem leaf powder, respectively, in 100 mL of distilled water overnight, followed by filtration through a cheesecloth to remove the coarse leaf materials. The filtered extract was centrifuged at 5000 rpm for 5 m, to remove the remaining particles and obtain a clear extract.

### 5.2. Preparation of Azadirachtin (NeemAzal T/S) (AZA)

A working concentration of 0.01% and 0.1% were prepared of NeemAzal T/S obtained from Trifolio Gmbh, Germany, containing (1% azadirachtin), a registered plant protection commercial product in the European Union, by dissolving 1 mL and 10 mL NeemAzal T/S in 100 mL of distilled water, respectively.

### 5.3. Preparation of Mefenoxam (MEF)

To prepare Mefenoxam as a positive control, Apron XL 350 FS (350 g/L Mefenoxam, Syngenta AG, Switzerland) was prepared as per the EU registered rate (3 mg/kg seeds) by homogeneously coating the seeds and keeping them at room temperature until they were dried.

### 5.4. HPLC Analysis of Neem Leaf Extract

To determine the azadirachtin content in the leaves of neem plant, an HPLC analysis was conducted in the Food Analysis laboratory of the Szent István University Gödöllő, Hungary. Five grams of ground leaves were extracted by shaking for 15 min with 100 mL HPLC grade methanol or water followed by subjection to ultrasonication in a water bath ultrasonic device (Raypa Model UCD-150) at a maximum frequency of v = 230 and W = 450 for 5 min. The mixture was stored overnight in a refrigerator. The supernatant was first filtered through filter paper and finally through a 0.22 µm, 25 mm hydrophilic PTFE syringe filter, before injection onto HPLC instrument. For the standard (Std), azadirachtin A (>95% pure, Sigma Aldrich, St. Louis, MO, USA) was used for comparison.

The HPLC runs and data processing were operated by EZChrom Elite software; external standard solution of 250 µg/mL in methanol was used for the quantitative determination of azadirachtin and their possible derivatives. Peak identification was based on comparing retention time and spectral characteristics with those of standard material.

In order to find the factor, which is needed to calculate the azadirachtin content, the following formula was used:
**250** (Std azadirachtin (µg/mL)/**5.5** (Area of std peak from chromatogram) × **100** (Total volume system mL)/**5** (weight of leaves in grams) = **909** (Factor)

The amount of azadirachtin A and the other peaks (which are suspected to be the derivatives or isomers of azadirachtin) in the leaf extracts were calculated by multiplying the area of the peaks with the factor. The final amount was calculated in mg/5 g neem leaves. 

### 5.5. Pathotype Characterization of the Tested P. halstedii Isolate

Sunflower leaves infected by *Plasmopara halstedii* were collected in the Jász-Nagykun-Szolnok county of Hungary. The isolate was stored at −70 °C in a deep freezer. Preparation of inoculum and pathotype characterization of the isolate was performed by the internationally standardized method of Trojanová et al. [36] and Bán et al. [12]. Nine sunflower differential lines containing different resistance genes to *P. halstedii* were used (Table 4) (Iregi szürke csíkos is a Hungarian sunflower cultivar with no resistance genes to *P. halstedii*). Seeds of each line (20 seeds/lines) were surface sterilized and germinated between wet filter papers at 21 °C for 3 days. Sporangia from infected sunflower leaves were washed off in bidistilled water. The concentration of inoculum was measured by hemocytometer and adjusted to 50,000 sporangia/mL. The whole seedling immersion (WSI) method by Cohen and Sackston [8] was used for inoculation. Inoculated seedlings (incubated in sporangial suspension at 16 °C in dark for 1 day) were sown in a given order (as listed in Table 4) in trays containing perlite (d = 4 mm). Plants were grown in a phytotron at 21 °C with a photoperiod of 12 h. Ten days after inoculation, plants were sprayed with bidistilled water, covered by plastic bags, and kept at 19 °C in dark overnight to induce sporulation. The disease was evaluated firstly after sporulation, according to the white sporangial coating on cotyledons, and secondly, based on damping-off, as well as according to the symptoms (chlorosis) on true leaves of 21-day old plants. Reaction of plants was determined as susceptible (S) or resistant (R), according to the result of the second evaluation. A score for each differential line was determined based on the reaction of the plants (S or R) and the location of the differential line inside the triplet: 1, 2, and 4 scores can be given for susceptible lines located in the first, second, and third place inside the triplet, respectively. The pathotype code was determined as the sum of scores by each triplet. The test was repeated twice with two repetitions by each.

### 5.6. P. halstedii Sporangia Collection and Preparation of Inoculum

The *P. halstedii* isolate used for this study was previously collected from Rákóczifalva, Hungary, in 2012. The isolate was stored at −70 °C in a deep freezer at the Plant Protection Institute of Szent István University (Gödöllő, Hungary) and identified as pathotype 704 according to the international method for pathotype identification by Trojanová et al. [36]. Detailed procedure of inoculum preparation is mentioned in the *in vitro* and *in vivo* experiment. 

### 5.7. In Vitro Experiment: Examination of the Effect of Neem-Derived Pesticides on P. halstedii Sporangial Germination

Infected sunflower leaves stored in deep freezer were soaked in 30 mL double distilled water to release the sporangia. One milliliter of sporangia suspension was diluted/mixed with 1 mL of each tested concentrations of neem leaf extract or azadirachtin solutions, and with 1 mL of Mefenoxam for positive control in an Eppendorf tube. It was agitated gently to mix uniformly and avoid bursting of sporangia, and was incubated at 16 ± 1 °C for 24 h in the dark in a thermostat. After a 24 h incubation period, samples were observed with a microscope at 200× magnification, to check the effect of neem derived pesticides on the sporangia morphology and release of zoosporangia. 

Microscopic examination was done for each tested treatment by counting first 50 sporangia/treatment. The experiment was replicated five times with each treatment. Microscopic examination of sporangia in bidistilled water (BW) served as a negative control. Based on the microscopic examination, we establish a germination scale (from 0–2) to identify the morphology of sporangia, wherein, 0 = Completely full sporangia, 1 = Partial empty sporangia, 2 = Completely empty sporangia. This scale is built and developed on the hypothesis that every single released zoospore is capable of infecting the host plant.

### 5.8. In Vivo Experiment: Pre- and Post-Treatment Effect of Neem-Derived Pesticides on P. halstedii in Sunflower

#### 5.8.1. Pre-Treatment Effect of Neem-Derived Pesticides Against *P. halstedii* in Sunflower

The Whole Seedling Immersion (WSI) method, Cohen and Sackston [8], was used for this experiment. The following 12 treatments were used in the pre-treatment experiment: Seedlings inoculated with *Plasmopara halstedii* sporangial suspension.Seedlings treated with bidistilled water (BW).Treated seeds (Mefenoxam) inoculated with *Plasmopara halstedii* sporangial solution.Treated seeds (Mefenoxam) inoculated with bidistilled water (BW).Seedlings pre-treated with AZA 0.1% inoculated with *Plasmopara halstedii* sporangial solution.Seedlings pre-treated with AZA 0.1% inoculated with bidistilled water (BW).Seedlings pre-treated with AZA 0.01% inoculated with *Plasmopara halstedii* sporangial solution.Seedlings pre-treated with AZA 0.01% inoculated with bidistilled water (BW).Seedlings pre-treated with NLE 10% inoculated with *Plasmopara halstedii* sporangial solution.Seedlings pre-treated with NLE 10% inoculated with bidistilled water (BW).Seeds pre-treated with NLE 20% inoculated with *Plasmopara halstedii* sporangial solution.Seedlings pre-treated with NLE 20% inoculated with double distilled water (BW).

Three-day old germinated seedlings of susceptible sunflower var. Iregi szürke csíkos (25 seedlings) were firstly immersed in NeemAzal T/S (0.1%, 0.01%) or neem leaf extract (10%, 20%) solutions, respectively, for 2 h. These treated seedlings were further inoculated with *P. halstedii* by immersing them in the sporangial suspension, which was adjusted to 50,000 sporangia/mL using a hemocytometer, and then incubated at 16 °C overnight in a dark place. For negative control, germinated seeds were first immersed in different treatments for 2 h, followed by immersion in bidistilled water (BW) for 24 h. Germinated seeds were planted in pots placed in a tray containing the moistened perlite with 5 seeds/pot, and with the five repetitions, and placed in the growth chamber with the controlled conditions (22 °C, with the photoperiod of 12 h, Relative humidity = 60%). Seedlings were watered regularly for 10 days. After 10 days, when the plants developed true leaves of about 1 mm in length, bidistilled water was sprayed using a hand sprayer on the plant leaves, enclosed in trays with a lid, covered in a dark polyethylene bag (to saturate it with moisture), and kept overnight at 19 °C under completely dark conditions to induce sporulation. The next day after sporulation, the first evaluation was done based on the cotyledons bearing sporangia (white growth). Plant growth characteristics, such as height, was measured as well. Plants were kept back in the growth chamber at 22 °C, with the photoperiod of 12 h, RH = 60%, and watered regularly. After 19 days post-sowing, the second evaluation, i.e., presence or absence of chlorosis, damping-off of seedlings, was done and recorded.

#### 5.8.2. Post-treatment Effect of Neem-Derived Pesticides Against *P. halstedii* in Sunflower 

The Whole Seedling Immersion method, Cohen and Sackston [8], was used for this experiment. The following 12 treatments were used in the post-treatment experiment:Seeds inoculated with *Plasmopara halstedii* sporangial suspension.Seeds inoculated with bidistilled water (BW).Pre-treated seeds (Mefenoxam) inoculated with *Plasmopara halstedii* sporangial solution.Pre-treated seeds (Mefenoxam) inoculated with bidistilled water (BW).Seeds inoculated with *Plasmopara halstedii* sporangial solution followed by AZA 0.1% solution.Seeds inoculated with bidistilled water followed by AZA 0.1% solution (BW).Seeds inoculated with *Plasmopara halstedii* sporangial solution followed by AZA 0.01% solution.Seeds inoculated with bidistilled water followed by AZA 0.01% solution (BW).Seeds inoculated with *Plasmopara halstedii* sporangial solution, followed by NLE 10% solution.Seeds inoculated with bidistilled water followed by NLE 10% solution (BW).Seeds inoculated with *Plasmopara halstedii* sporangial solution followed by NLE 20% solution.Seeds inoculated with bidistilled water followed by NLE 20% solution (BW).

Three-day old germinated seedlings of susceptible sunflower var. Iregi szürke csíkos were first inoculated with *Plasmopara halstedii* sporangial suspension for 24 h, which was adjusted to 50,000 sporangia/mL using a hemocytometer, followed by respective treatments, for 2 h. For negative control, germinated seedlings were first treated in bidistilled water (BW) for 24 h followed by respective treatments for 2 h. Germinated seedlings were planted in pots placed in a tray containing the moistened perlite with 5 seeds/pot, and with the five repetitions, and placed in the growth chamber with the controlled conditions (22 °C, with the photoperiod of 12 h, RH = 60%). Plants were watered regularly. After 10 days of sowing, when the plants developed true leaves of about 1 mm, bidistilled water was sprayed onto the seedlings and enclosed in trays with a lid, covered in a dark polyethylene bag (to saturate it with moisture), and kept overnight at 19 °C under completely dark conditions to induce sporulation. The next day after sporulation, first evaluation was done based on the cotyledons bearing sporangia. Plant growth characteristics, such as height, was measured as well. Plants were kept back in the growth chamber at 22 °C, with the photoperiod of 12 h RH = 60% and watered regularly. After 19 days of sowing, a second evaluation, i.e., presence or absence of chlorosis on true leaves, damping-off of seedlings, was done and recorded.

### 5.9. Data Analysis

For both, the *in vitro* and for the *in vivo* experiments, ANOVA followed by a post-hoc Tukey test was performed to compare the different treatments in R software v 3.4.0 R Core Team [37], while graphs were made in Excel.

## Figures and Tables

**Figure 1 plants-09-00535-f001:**
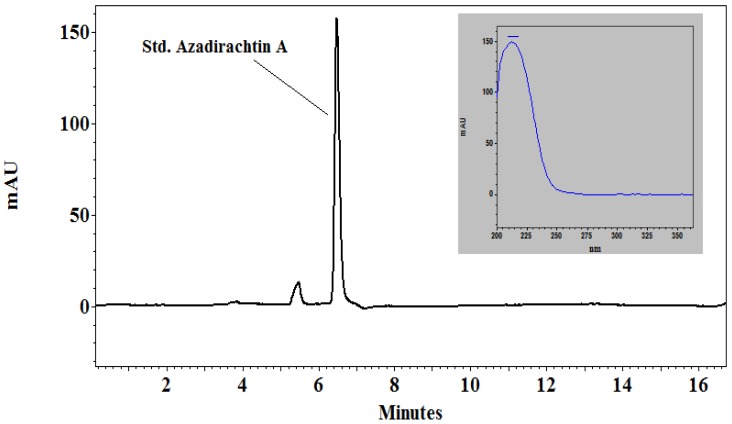
HPLC chromatogram of standard azadirachtin A.

**Figure 2 plants-09-00535-f002:**
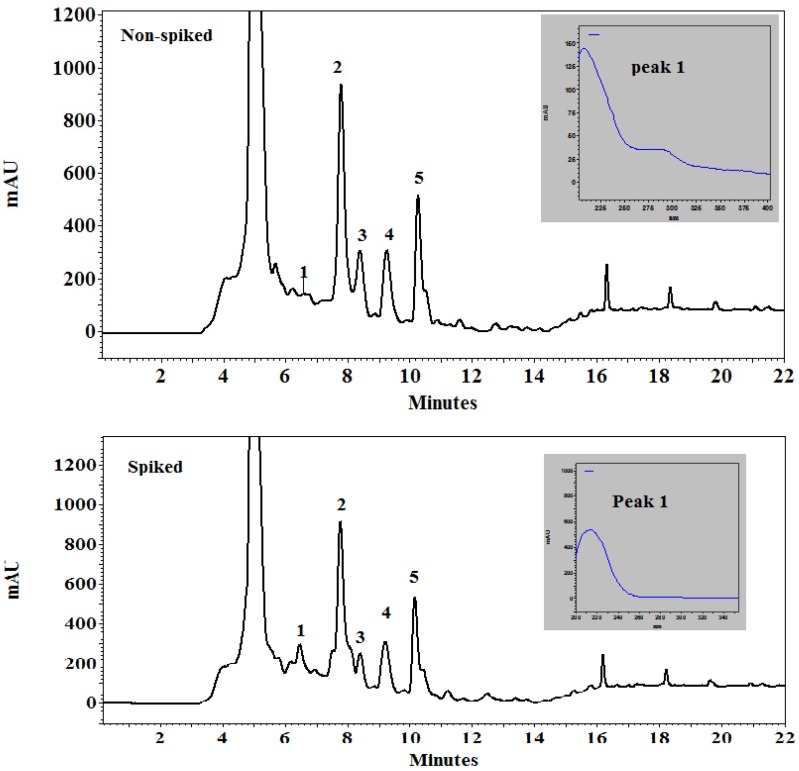
HPLC chromatogram of neem leaf extract. The top figure is “non-spiked”, which means standard azadirachtin A solution was not added externally. The bottom figure is “spiked”, which means standard azadirachtin A was added externally in the neem leaf extract sample before performing the test.

**Figure 3 plants-09-00535-f003:**
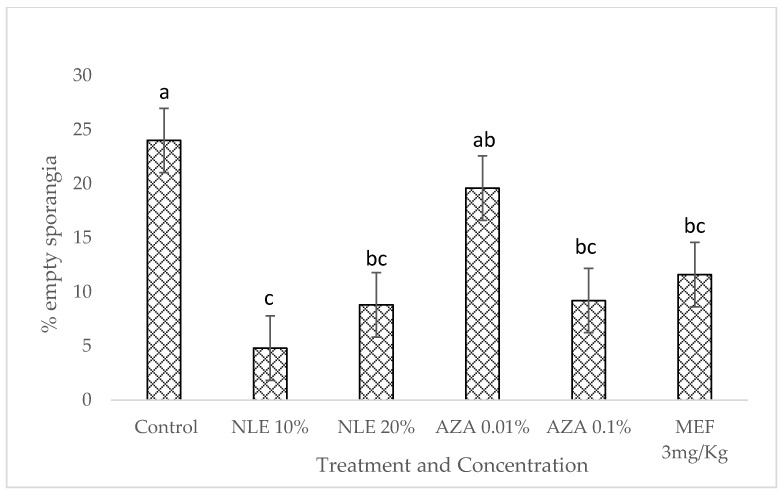
Effect of two different concentrations of neem leaf extract (NLE) and NeemAzal T/S (AZA), respectively, on the germination of *P. halstedii* sporangia. Mefenoxam (MEF) was used as a positive control). Different letters according to Tukey’s test indicate significant difference at 95% confidence level.

**Figure 4 plants-09-00535-f004:**
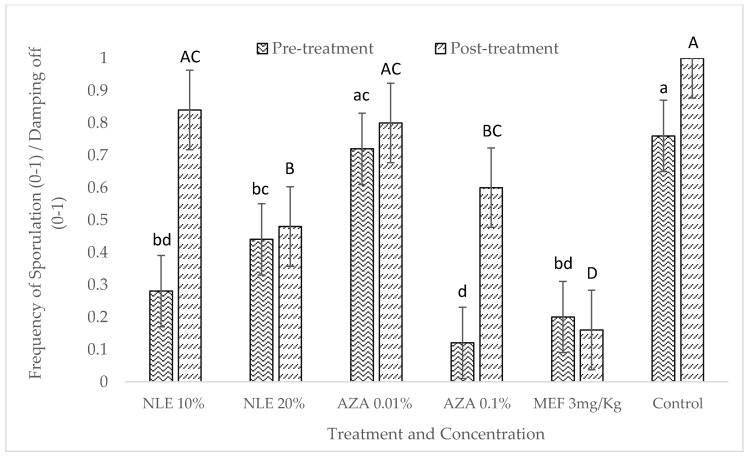
Pre- and Post-treatment effects of neem leaf extract (NLE) and NeemAzal T/S (AZA) on *P. halstedii* sporulation in susceptible sunflower seedlings. Different lowercase letters represent significant difference comparing the pre-treatment effect. Different uppercase letters represent significant difference to compare post-treatment effect according to Tukey’s test at 95% confidence level.

**Figure 5 plants-09-00535-f005:**
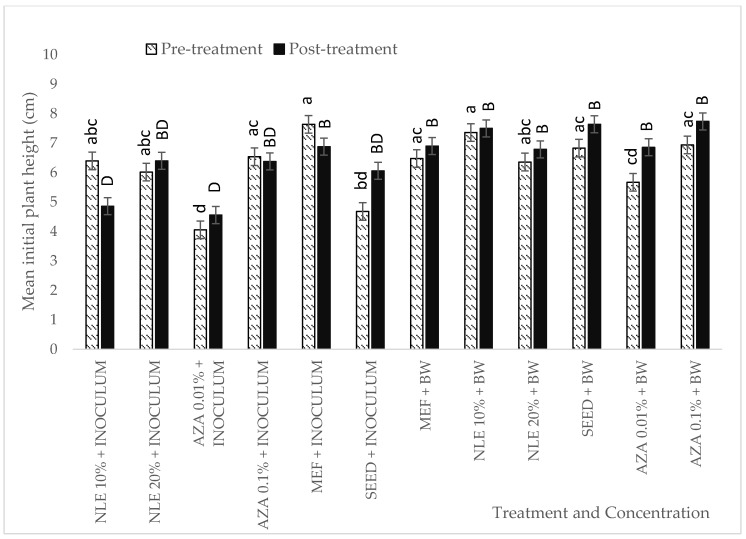
Measurement of initial plant height of seedlings pre- and post-treated with different concentrations of neem leaf extract (NLE) and NeemAzal T/S (AZA), and with bidistilled water (BW) serving as negative control, as a part of first evaluation. ANOVA post-hoc Tukey test was performed on the data. Different lowercase letters represent significant difference comparing the pre-treatment effect. Different uppercase letters represent significant difference to compare post-treatment effect according to Tukey’s test at 95% confidence level.

**Table 1 plants-09-00535-t001:** Table representing the area of the peaks as analyzed from the chromatogram and calculating the amount of azadirachtin present in the given neem leaf extract samples. ** is the azadirachtin A concentration found in the neem leaf extracts.

Sample	Peak No	Area (From Chromatogram)	Area × 909 (Factor)	Azadirachtin (µg/g)	Azadirachtin (mg/5 g)
Non-Spiked	1	0.23		209	1 **
	2	47.2		42,909	214.5
	3	15.5		14,089	70
	4	18.06		16,425	80
	5	18.6		16,907	84.5
Spiked	1	5.5		4999	24.5 **
	2	37.7		34,269	171
	3	6.5		5908	29.5
	4	17.5		15,907	79.5
	5	18.4		16,725	83.5

**Table 2 plants-09-00535-t002:** Pathotype characterization of *P. halstedii* isolate used in the test. (S = Susceptible, R = Resistant).

Differential Lines	First Evaluation (%)	Second Evaluation (%)	Reaction of Plants	Score	Pathotype Code
Iregi szürke csíkos	96.7± 3.9	100	S	1	
RHA-265	91.7 ± 6.4	100	S	2	7
RHA-274	100	100	S	4	
PMI-3	0	0	R	0	
PM-17	0	0	R	0	0
803-1	0	0	R	0	
HAR-4	0	0	R	0	
QHP-2	0	0	R	0	4
HA-335	93.3 ± 5.4	100	S	4	

**Table 3 plants-09-00535-t003:** Table showing ANOVA results for both pre- and post-treatment effect of *P. halstedii* sporulation under *in vivo* conditions with *p* value, significantly different at 95% confidence interval. (Df = Degrees of freedom. Sq = square)

Treatment	Df	Sum Sq	Mean Sq	F Value	*p* Value
Pre-treatment	11	22.33	2.03	21.31	<0.05
Post-treatment	11	42.68	3.88	48.66	<0.05

**Table 4 plants-09-00535-t004:** Sunflower differential lines used for pathotype identification for *P. halstedii* in the experiment and resistance genes incorporated (based on Gascuel et al. [10]).

Sunflower Differential Line	Iregi Szürke Csíkos	RHA-265	RHA-274	PMI-3	PM-17	803-1	HAR-4	QHP-2	HA-335
Resistance gene to *P. halstedii*	No *Pl* gene	*Pl1*	*Pl2/Pl21*	*Pl_PMI3_*	*Pl5*	*Pl5+*	*Pl_15_*	*Pl1/Pl_15_*	*Pl6*

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
