# Peer review of "Preliminary Investigation of Effect of Neem-Derived Pesticides on Plasmopara halstedii Pathotype 704 in Sunflower under In Vitro and In Vivo Conditions"

_plants, 2020, doi:10.3390/plants9040535_

Round 1

Reviewer 1 Report

REVIEW – Doshi et al.

My two main comments:

1) additional background information on the problem (specific to sunflower) could be included in introduction -- some of this is already in the discussion and can be moved up to the intro and further developed.

2) improvements can still be made in how the results are presented. Some results are in tables that should be in the text (ANOVA results), and others  could be presented in tables (Turkey comparisons).

Abstract: suggested rewrite “Two neem-derived pesticides were examined under in vitro and in vivo conditions to test their efficacy in controlling Plasmopara halstedii pathotype 704, a causal agent of downy mildew in sunflower.”

Line 64: lines repeated twice

Table 2 and 3 show the results of ANOVAs, but this information should be included in the text and the results of the Tukey pairwise comparisons (estimate, SE, z-ratio, p-value) presented in the tables. Also, if you want to use symbols to communicate significance of p-values, then remove the actual p-value from the table and include in table legend or footnote what the symbols mean (‘*’ significant at p < 0.05, ‘**’ significant at p < 0.01, ‘***’ significant at p < 0.001).

Line 113: change to “was not significantly different from the infected control (Seed + Inoculum).”

Line 125: insert space between “shorter plant”

Discussion: First paragraph should be moved up to introduction (insert at line 55) to provide background on the problem being examined in the present study.

Line 207: change to “ground into powder”

Line 210: remove “for”

Line 216: change to “in 100 mL of distilled water”

Line 228: change “lines” to line

Line 272-273: remove “as the followings”

Author Response

Dear Reviewer 1,

Thank you for reviewing our manuscript and for your comments and suggestions. We have also included new data and information in the manuscript. Please find the attached responses to the comments and suggestions in the manuscript (using track changes).

I hope you will accept the responses and accept our manuscript for publication.

Yours Sincerely,

Pratik Doshi

----------------------------------------------------------------------------

REVIEW – Doshi et al.

My two main comments:

  • additional background information on the problem (specific to sunflower) could be included in introduction -- some of this is already in the discussion and can be moved up to the intro and further developed.

We have included additional background information on the problem (specific to sunflower) in the introduction. These changes are highlighted using Track changes.

2) improvements can still be made in how the results are presented. Some results are in tables that should be in the text (ANOVA results), and others could be presented in tables (Turkey comparisons).

We improved the results represented in the manuscript. We have included new results regarding HPLC analysis of the neem leaf extract to measure the azadirachtin concentration in the chosen working concentrations. Results are interpreted in the text.

Abstract: suggested rewrite “Two neem-derived pesticides were examined under in vitro and in vivo conditions to test their efficacy in controlling Plasmopara halstedii pathotype 704, a causal agent of downy mildew in sunflower.”

The change has been included in the abstract using track changes.

Line 64: lines repeated twice

Deleted “lines” which was repeated.

Table 2 and 3 show the results of ANOVAs, but this information should be included in the text and the results of the Tukey pairwise comparisons (estimate, SE, z-ratio, p-value) presented in the tables. Also, if you want to use symbols to communicate significance of p-values, then remove the actual p-value from the table and include in table legend or footnote what the symbols mean (‘*’ significant at p < 0.05, ‘**’ significant at p < 0.01, ‘***’ significant at p < 0.001).

The results of ANOVA and Tukey test has been interpreted in the text. Significant difference has been shown as p<0.05.

Line 113: change to “was not significantly different from the infected control (Seed + Inoculum).”

Changed at line 156 using track changes.

Line 125: insert space between “shorter plant”

Inserted space between shorter plants at line 168 using track changes

Discussion: First paragraph should be moved up to introduction (insert at line 55) to provide background on the problem being examined in the present study.

First paragraph moved up to the introduction.

Line 207: change to “ground into powder”

Change performed using track changes

Line 210: remove “for”

Change performed using track changes

Line 216: change to “in 100 mL of distilled water”

Change performed using track changes

Line 228: change “lines” to line

Change performed using track changes

Line 272-273: remove “as the followings”

Change performed using track changes

Reviewer 2 Report

Dear Authors, you should address my recommendations highlighted inside the text and Figures.

Author Response

Dear Reviewer 2,

Thank you for reviewing our manuscript and for your comments and suggestions. We have also included new data and information in the manuscript. Please find the attached responses to the comments and suggestions in the manuscript (using track changes).

I hope you will accept the responses and accept our manuscript for publication.

Yours Sincerely,

Pratik Doshi

Reviewer 3 Report

Dear authors,
you have carried out an interesting and original research in order to evaluate the action of pesticides derived from neem, for example extracts of neem leaves and NeemAzal T / S on Plasmopara halstedii in vitro and in vivo conditions in sunflower. The use of extracts from Neem could be a valid tool in the control of new aggressive pathotypes, such as the 704 pathotype, which have been able to overcome the effect of the Pl6 resistance gene against sunflower downy mildew.

However, the manuscript presents a strong criticality. In view of the fact that, the aqueous extract of Neem leaves contains bioactive secondary metabolites, as shown by the scientific literature, the reader would have expected in this study that the authors would face a phytochemical screening of aqueous extract of Azadirachta indica leaves. In this way you are not aware of what the aqueous extract, which you have obtained in your experimentation, contains. To use the data from the aqueous extract of Azadirachta indica leaves it is necessary to perform a phytochemical screening. I am convinced that you have this important data. If this were not the case, the data from the use of the Azadirachta indica leaf aqueous extract cannot confirm that the latter's action is exclusively due to the Neem.

I think you should improve this part of your work, so that the manuscript can be published in the Plants Journal.

Author Response

Dear Reviewer 3,

Thank you for reviewing our manuscript and for your comments and suggestions. We have also included new data and information in the manuscript. Please find the attached responses to the comments and suggestions in the manuscript (using track changes) and in italics + black font in the documents attached.

I hope you will accept the responses and accept our manuscript for publication.

Yours Sincerely,

Pratik Doshi

Round 2

Reviewer 3 Report

Dear Authors,

the new version of the manuscript, in which you made the suggestions proposed by the reviewers, appears significantly improved.

You should only make a small correction.

  • In line 55 enter only the numerical references of the authors you cited.
  • Delete Table 3 and enter the data obtained from the statistical analysis directly in the text.

The version of the manuscript presented now, with the  suggestions reported above, can be accepted for publication in the Plants Journal.

Author Response

Dear Reviewer 3,

Thank you for your comments and suggestions. Please find below the responses to your comments (in italics).

Kind Regards,

Pratik Doshi

(PhD Candidate, Corresponding author)

----------------------------------------------------------------------------

In line 55 enter only the numerical references of the authors you cited.

The authors names are removed and only the numerical references are mentioned.

Delete Table 3 and enter the data obtained from the statistical analysis directly in the text.

Table 3 has been deleted and the data obtained from the statistical analysis has been mentioned in the text in lines 115-117.

This manuscript is a resubmission of an earlier submission. The following is a list of the peer review reports and author responses from that submission.

Round 1

Reviewer 1 Report

Dear Authors, you addressed my recommendations and, therefore, your manuscript can be accepted for publication in this Journal in my opinion.

Reviewer 2 Report

This manuscript reports partly interesting data for control of an important fungus using a natural product. However, many papers have already reported that need-derived compounds have biocidal properties and it is difficult to find novelty and scientific insight. The authors describe no previous records about the effects of neem-derived pesticides on P. halstedii in sunflower, but this is rather weak. In particular, the authors discuss systemic effect of neem, but no data are provided. Figure 3 may be the most important experiment, but this is not repeated in time, indicating less strong scientific evidence. The reviewer judges that this manuscript is still in a preliminary stage, as the authors mention in the title, and thus not suitable for publication in the journal.

Reviewer 3 Report

REVIEW – Doshi et al.

This is a comprehensive and interesting study examining the potential of neem-derived pesticides to control P. halstedii fungal infection in sunflowers. The manuscript is overall well-presented. I have just a couple major issues regarding the presentation of the results which need to be addressed before publication. This will be an important and timely contribution to the sunflower industry, as well as other agricultural commodity crops which could benefit from investigation of neem-derived pesticides as a more environmentally friendly alternative to chemical pesticides.

MAJOR COMMENTS

From the methods it appears 25 seedlings were used per treatment, but error bars showing variation among seedlings in each treatment were not included in the graphs. These must be added to show a measure of variation (either se or sd).

I would also like to see a table with the actual results from the ANOVAs (including statistics, df, and p-values).

MINOR COMMENTS

L35-37: rewrite for clarity

L65-68:  suggested rewrite “All the treatments except AZA 0.01% were found to be significantly better than the control (no treatment applied) at reducing the total number of empty sporangia (which includes completely empty and partially empty sporangia as per the proposed scale; Fig. 1). Both concentrations of neem leaf extract and the highest concentration of NeemAzal T/S gave similar results to those of mefenoxam”.

Figure 1: Error bars showing variation? Title can be removed from figure.

L96: change to “significantly shorter”

L97: change to “not significantly different from the infected control”

Figure 3 caption: Define BW.

L108: change to “significantly shorter plant height”

L111: halstedii is misspelled

L126: remove comma and semi-colon, put (known for antifungal properties) in parentheses

L135: change resulted to suggested

L152-155: rewrite for clarity

L181: change grounded to ground

L190: change semi-colon to comma and change to “in 10 mL of distilled water, respectively.”

L194: change kept to keeping

L215: change to “is capable of infecting”

L220: change to “The following 12 treatments were used in the pre-treatment experiment:”

L222-232: change seeds to seedlings, since the treatments were done on 3-day old seedlings

L242: remove “of plantation”

L243: 1 mm in length or width?

L249: change "of plantation" to post-sowing

L56-257: This is the same as listed in the pre-treatment. Were seeds treated with Mefenoxam after innoculation with P. halstedii for the post-treatment?

L284: change to “ANOVA followed by post-hoc Tukey tests were performed”

L285: Tests for normality and equal variance? Any data transformations? Need to include package used and R citation.

Reviewer 4 Report

Althought the manuscript reports interesting results about the potential use of neem-derived pesticides on Plasmopara halstedii, I believe that the data reported are poor and the article is not sustainable for pubblication in the current form. The main flaw is reported below

each experiment must be repeated because additional experiments are needed to demonstrate for the first time the activity of neem-derived pesticides against the pathogen.